# scKGOT: Intercellular Signaling Inference with Knowledge Graph Optimal Transport

## Abstract

Single-cell transcriptomics provides detailed genetic insights into cellular heterogeneity within intact organs and the intercellular signaling that underpins tissue homeostasis, development, and disease. To improve the inference of intercellular signaling and pathway activity, we introduce scKGOT, a novel method that employs the Knowledge Graph Optimal Transport (KGOT) algorithm to model and quantify ligand-receptor-signaling networks between sender and receiver cells. scKGOT defines sender and receiver spaces using pairwise distance matrices from gene expression profiles and leverages prior knowledge from the Ligand-Receptor-Pathway Knowledge Graph (LRP-KG) as initial guidance for transport optimization, allowing for dynamic adaptation based on gene expression data. Through comprehensive benchmarking on public single-cell transcriptomic datasets, scKGOT consistently outperforms existing inference methods in terms of precision and interpretability. Furthermore, we demonstrate its practical applicability across multiple case studies, uncovering complex pathway interactions and revealing insights into cellular heterogeneity in diverse biological contexts. By incorporating scKGOT, we provide a robust and generalizable approach for pathway inference in single-cell analyses, advancing the understanding of intercellular communication mechanisms and offering valuable insights into biological processes at the cellular level.

## 1 Introduction

Single-cell RNA sequencing (scRNA-seq) technologies are increasingly being used to characterize the heterogeneity of a complex tissue (Macosko et al., 2015; Klein et al., 2015). Beyond annotating cell types and transcript abundance, it is important to understand the underlying mechanism of cell-cell communication within the tissue microenvironment (Shao et al., 2020; Armingol et al., 2021). scRNA-seq technology holds great promise for investigating cell-cell communication mediated by ligand-receptor interactions at gene expression level. Several methods have been developed to infer ligand-receptor pairs that are active between two cell types (Efremova et al., 2020; Jin et al., 2021; Cheng et al., 2021). They focus on direct predictions of ligand-receptor pairs based on gene expression and the correlation between genes.

Due to the intertwined nature of biological pathways, simply examining expression levels of ligand and receptor genes cannot reliably capture the activated signaling pathway mediating intercellular communication. As a step forward, NicheNet (Browaeys et al., 2020) and CellCall (Zhang et al., 2021) aim to identify both ligand-receptor pairs and genes downstream of them. However, existing methods hardly make full use of biological pathways to infer cell-cell communication for the fact holds that ligand-receptor-mediated cell-cell communication relies on the activation of the specific signaling pathways, e.g., JAK-STAT pathway, PKC pathway, and MAPK pathway (Hu et al., 2021). It is still a great challenge to accurately model the intercellular ligand-receptor signaling pathways for the inference of cell-cell communication.

Inspired by the remarkable performance of optimal transport in numerous tasks (Cang & Nie, 2020; Cang et al., 2023), we herein introduce single-cell Knowledge Graph Optimal

Transport (scKGOT), a method for construction of a signaling network (union of multiple signaling pathways) to infer ligand-receptor-mediated cell-cell communication using our proposed scKGOT algorithm for the first time based on single-cell transcriptomic data. The algorithm first equips with a Ligand-Receptor-Pathway Knowledge Graph (LRP-KG) consisting of intra-cellular and intercellular functional gene-gene interactions with different types of pathways. It then identifies the activated signaling pathways by addressing a novel scKGOT problem that incorporates gene importance and pathway Knowledge Discrepancy (KD), highlighting highly confident ligand-receptor pairs and top-ranked pathways. Pathways involving ligand-receptor-mediated cell-cell communications are then reconstructed based on predictions and known facts in LRP-KG.

Inspired by the remarkable performance of optimal transport in various tasks (Cang & Nie, 2020; Cang et al., 2023), we introduce single-cell Knowledge Graph Optimal Transport (scKGOT), a method for constructing signaling networks (unions of multiple signaling pathways) to infer ligand-receptor-mediated cell-cell communication from single-cell transcriptomic data. Our scKGOT algorithm is equipped with a Ligand-Receptor-Pathway Knowledge Graph (LRP-KG), which encompasses both intra-cellular and intercellular gene-gene interactions across different types of pathways. The algorithm identifies activated signaling pathways by solving a novel scKGOT problem that incorporates gene importance and pathway Knowledge Discrepancy (KD), focusing on highly confident ligand-receptor pairs and top-ranked pathways. Subsequently, it reconstructs pathways involving ligand-receptor-mediated cell-cell communications based on both predictions and existing knowledge within the LRP-KG.

The primary objective of the scKGOT algorithm is to find an optimal transportation plan in pathways from ligands to receptors based on LRP-KG, considering genes from different pathways, cell-type–specific expression, and correlation connection to highly active ligand-receptor pairs. We hypothesize that by integrating prior knowledge of signaling pathways and modeling the fine-grained gene interactions, scKGOT will outperform existing methods in terms of precision and interpretability. To test this hypothesis, we benchmark the performance of scKGOT using carefully curated scRNA-seq datasets with ground truth ligand-receptor-mediated cell-cell communication across 11 human and mouse tissues.

To demonstrate scKGOT's utility in uncovering biological insights, we conducted a multi-level analysis of pathway activation, showcasing its capability to explore complex signaling dynamics. We also performed a pathway interaction analysis of cell type pairs, including individual analyses of specific cell type pairs and comparisons between tumor and non-tumor cell types, further revealing scKGOT's potential to uncover key biological insights.

## 2   PROBLEM FORMULATION

Previous works (Browaeys et al., 2020; Efremova et al., 2020) have shown that the binding problem of ligand-receptor pairs can be formulated as a classification task from a probabilistic perspective, which predicts the existence of a binding between ligand and receptor genes given a dataset $D$.

$$\hat{z} = \arg\max_{z \in \mathcal{C}} P(z \mid D) \tag{1}$$

where $\mathcal{C}$ is the candidate set of interest containing ligand-receptor pairs.

From a fine-grained deconstruction perspective, we propose a novel formulation that considers the underlying transportation of signals through multiple pathways. Denoting the space of pathways as $\mathcal{W}$, we formulate the signaling transportation between ligands and receptors across pathways as follows:

$$\begin{aligned}
\hat{z} &= \arg\max_{z \in \mathcal{C}} \Sigma_{w_n \in \mathcal{W}} \ P(z \mid w_n, D) \cdot P(w_n \mid D) \\
&= \arg\max_{z \in \mathcal{C}} \Sigma_{w_n \in \mathcal{W}} \ s_1(z, w_n, D) \cdot s_2(w_n, D)
\end{aligned} \tag{2}$$

Here, $w_n$ represents a pathway retrieved from the LRP-KG space of pathways $\mathcal{W}$. In this factorization, the first factor $P(z \mid w_n, D)$ corresponds to the gene importance score of ligand-receptor pairs within pathway $w_n$, while the second factor $P(w_n \mid D)$ reflects the pathway knowledge discrepancy (KD) for pathway $w_n$ given a dataset $D$. For each pair $\hat{z}$, the most relevant pathways are ranked by KD, with smaller values indicating better alignment.

This formulation (Eq. 2) provides several key advantages: (1) It retains the original probabilistic formulation, making our model comparable to previous research and machine learning baselines, ensuring fair benchmarking. (2) It explicitly models the transportation of ligand-receptor pairs through multiple pathways, offering additional insights for further analysis, such as identifying the dominant pathway behind active ligand-receptor pairs. (3) It can be extended to zero-shot scenarios, where candidate ligand-receptor pairs are not limited by hand-crafted priors, enabling the discovery of ligand-receptor pairs and pinpointing critical components of highly activated pathways.

## 3 Knowledge Graph Optimal Transport (KGOT)

To address the above problem in a fine-grained perspective, we developed the Knowledge Graph Optimal Transport (KGOT) framework for pairing ligands and receptors across pathways between two cell types. In scKGOT, we defines a sender space $(\mathbf{C}_1, \mathbf{p})$ and a receiver space $(\mathbf{C}_2, \mathbf{q})$, where $\mathbf{C}_1 \in \mathbb{R}^{a \times a}$ and $\mathbf{C}_2 \in \mathbb{R}^{b \times b}$ are pairwise distance matrices derived from the gene expression profiles of the sender and receiver cell types, respectively. The vectors $\mathbf{p} \in \mathbb{R}^a$ and $\mathbf{q} \in \mathbb{R}^b$ represent the marginal distributions of gene expression levels, capturing the relative abundance of each gene within the sender and receiver cells, respectively. Pathways from the LRP-KG are used as prior knowledge, providing initial estimates that guide the search process. However, the final optimal transport solution is primarily influenced by gene expression data, allowing scKGOT to adapt dynamically and discover new interactions that extend beyond the predefined pathways.

scKGOT simulates signal transmission by leveraging the LRP-KG as a foundation for signal transportation, integrating prior knowledge with data-driven insights to identify key ligand-receptor interactions and discover novel pathways. It uses single-cell datasets from various species and organs. For each cell type pair of interest, scKGOT predicts ligand-receptor relationships by enumerating multiple pathways and calculating the probability distribution of ligand-receptor pairs as a weighted average across signaling transportation problems. The model predicts two key components in this framework: (1) gene importance score, indicating the importance of gene pairs within specific pathways, and (2) pathway knowledge discrepancy (KD), estimated as the total cost of signal transportation across the pathways.

To obtain the optimal transport plan $\gamma^* \in \mathbb{R}^{a \times b}$, we minimize the loss function $\mathcal{L}(\mathbf{C}_1, \mathbf{C}_2, \gamma)$, defined as follows:

$$
\begin{aligned}
\gamma^* &= \arg\min_{\gamma} \mathcal{L}(\mathbf{C}_1, \mathbf{C}_2, \gamma) \\
&= \arg\min_{\gamma} \sum_{i,j,k,l} L\left(\mathbf{C}_{1i,j}, \mathbf{C}_{2k,l}\right) \cdot \gamma_{i,k} \cdot \gamma_{j,l} \\
\text{s.t.} \quad &\gamma \geq 0, \ \gamma \mathbf{1} \leq \mathbf{p}, \ \gamma^T \mathbf{1} \leq \mathbf{q}, \\
&\mathbf{1}^T \gamma \mathbf{1} = m \leq \min\left\{\|\mathbf{a}\|_1, \|\mathbf{b}\|_1\right\}.
\end{aligned}
\tag{3}
$$

where $L\left(\mathbf{C}_{1i,j}, \mathbf{C}_{2k,l}\right)$ represents the square loss between the correlation distance matrices $\mathbf{C}_1$ and $\mathbf{C}_2$, where $i$ and $j$ are genes from sender cells and $k$ and $l$ are genes from receiver cells. The variables $\mathbf{p}$ and $\mathbf{q}$ represent the marginal distributions of gene expression for the sender and receiver cells, respectively.

The optimal solution $\gamma^*$ directly computes $s_1$ and $s_2$. The gene importance score $s_1(z, w_n, D)$ is derived from the transport plan, reflecting how well ligand-receptor pairs align with the observed data in a specific pathway.

$$s_1(z, w_n, D) = \frac{\gamma_{i,k}^*}{\sum_{i,k} \gamma_{i,k}^*} \tag{4}$$

$\gamma_{i,k}^*$ represents the transport mass for the gene pair $(i, k)$, and the sum $\sum_{i,k} \gamma_{i,k}^*$ provides the normalization. Meanwhile, $s_2(w_n, D)$ is calculated as the normalized transport cost,

$$s_2(w_n, D) = norm(\mathcal{L}(\mathbf{C}_1, \mathbf{C}_2, \gamma^*)) \tag{5}$$

The function $norm(\mathcal{L}(\mathbf{C}_1, \mathbf{C}_2, \gamma^*))$, which converts the minimized loss into a maximization framework using $norm(\cdot) = 1 - \text{Percentile}(\cdot)$. This transformation ensures that a lower transport cost results in a higher $s_2$ value, representing a better alignment of the pathways with the dataset information. By maximizing $s_2$, scKGOT identifies pathways that are most consistent with the observed gene expression data, enabling the model to adapt and reveal new biological insights beyond the initial pathway information provided by the LRP-KG.

By deconstructing the LRP-KG into distinct signaling pathways, scKGOT provides an ensemble view of gene-level intercellular communication, emphasizing the complexity of biological signaling across various cell types. Its pathway-centric approach not only maps gene interactions across multiple routes but also highlights key genes with significant expression, using LRP-KG to decode complex cellular signaling networks and solve the KGOT problem.

scKGOT offers high interpretability, providing researchers with a profound understanding of inferred gene-level cell-cell communication networks. Its comprehensive suite of visualization tools—including heatmaps, Sankey diagrams, and network plots—enables researchers to dissect and scrutinize the computational predictions of gene importance scores and pathway knowledge discrepancies. These visual aids are crucial for identifying key genes and active pathways, elucidating their interconnections, and providing a deeper understanding of the complex mechanisms underlying intercellular communication.

## 4 EXPERIMENTS

### 4.1 EXPERIMENTAL SETTINGS

**Task and Baselines.** To evaluate the performance of scKGOT in predicting ligand-receptor binding, we frame this task as a multi-relation link prediction problem and compare scKGOT against several well-established knowledge graph embedding (KGE) methods which rely heavily on knowledge graphs, including TransE (Bordes et al., 2013), DistMult (Yang et al., 2014), RotatE (Sun et al., 2019), and ComplEx (Trouillon et al., 2016). These baseline methods account for various relational characteristics, such as symmetric, asymmetric, inverse, compositional, and 1-to-N relationships.

**Dataset.** We conducted experiments on 6 human and 5 mouse scRNA-seq datasets, with each dataset containing at least one pair of ligand and receptor within the cell pairs of interest. All scRNA-seq datasets were retrieved from several high-quality reports. For the construction of LRP-KG, KEGG (Kanehisa et al., 2017) and Reactome (Fabregat et al., 2018) provide 2,302 human pathways and 1,800 mouse pathways with 2,223,641 and 1,651,421 records of ligand-receptor interaction facts respectively, including binding, dephosphorylation and activation, etc.

**Metrics.** We report Mean Rank (MR) and Hits@$K$ ($K = 1, 5, 10, 50$) based on the filtered setting, which accounts for the varying sizes of rank candidates. The results, summarized in Fig. 1, are derived from five independent runs using different random seeds. For all experiments, we employ permutation testing with 100 iterations. For comparison with specialized cell-cell interaction prediction methods, we report accuracy and percentile rank.

### 4.2 PERFORMANCE COMPARISON

The evaluation protocol for ligand-receptor binding, framed as a multi-relation link prediction task, follows the standard practice of link prediction under the stochastic local closed

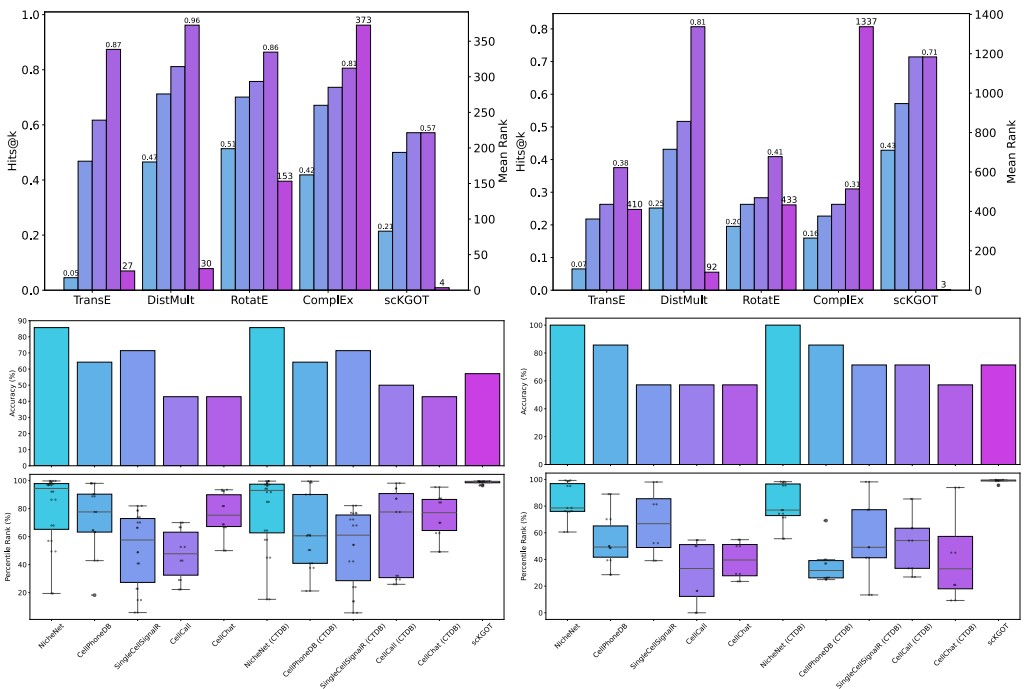

Figure 1: Performance comparison of different models for gene-gene interaction predictions.

world assumption, with a modification in the ensemble ranking procedure. Since ligand-receptor pairs may be involved in multiple pathways within our LRP-KG, we evaluate all ground truth triples and assign scores accordingly. The final rank for each ligand-receptor prediction is determined by the highest rank achieved among all scored triples, treating each pathway as a distinct relation in the evaluation process.

As shown in the upper section of Fig. 1, while Hits@$K$ metrics are comparable across methods, scKGOT significantly outperforms baseline models on Mean Rank by making predictions where both the LRP-KG and scRNA-seq data provide corroborating evidence, leading to more stable and accurate rankings compared to baseline models. This result demonstrates the effectiveness of integrating prior knowledge in LRP-KG with gene expression data.

To further validate scKGOT, we compared it with several baseline methods specifically designed for cell-cell interaction prediction, including NicheNet (Browaeys et al., 2020), CellPhoneDB (Efremova et al., 2020), SingleCellSignalR (Cabello-Aguilar et al., 2020), CellChat (Jin et al., 2021), and CellCall (Zhang et al., 2021). The results, shown in the lower section of Fig. 1, indicate that scKGOT achieves accuracy levels comparable to these baselines. However, scKGOT notably excels in percentile rank across all datasets, consistently ranking target ligand-receptor pairs within the top 1-5 positions out of hundreds or even thousands of potential candidates. This results in percentile rank values frequently approaching 0.999, indicating the model's ability to prioritize biologically relevant interactions with remarkable precision. The minimal variance observed in the box plots further underscores the robustness and reliability of scKGOT in producing consistently accurate top-ranked predictions.

One notable observation from our experiments is that while traditional methods perform well in many contexts, they may face challenges in scenarios that involve modeling signal propagation through multiple pathways. In contrast, explicit consideration of pathway complexity by scKGOT appears to provide a more nuanced understanding of the intricate dynamics of intercellular communication. Our results highlight the robustness of scKGOT in capturing the structural and functional relationships between genes. These findings underscore the potential of scKGOT as a powerful tool for gene-level intercellular communication analysis.

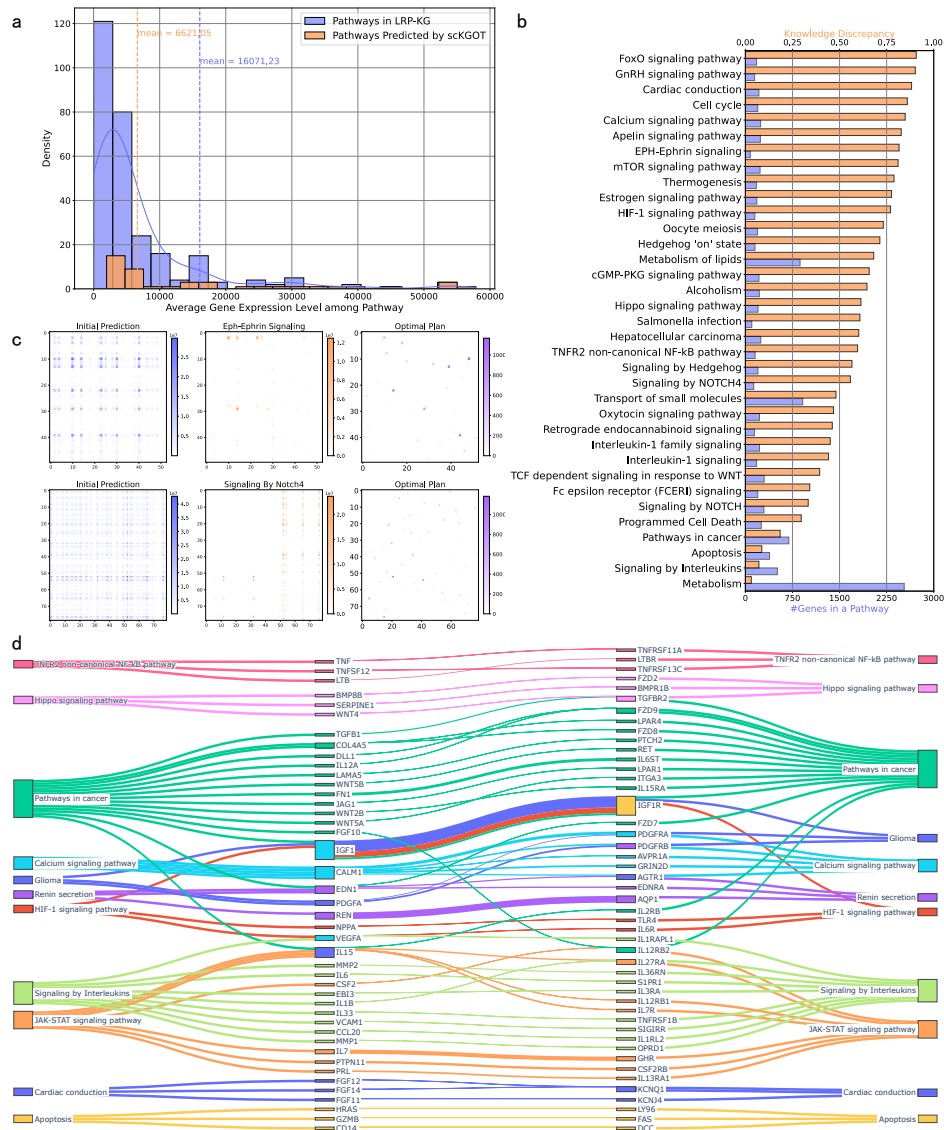

Figure 2: Structured methodology from cellular to pathway-level analysis. (a) Distributions of pathway knowledge discrepancy via Kernel Density Estimation (KDE). (b) Size-insensitivity in pathway identification, demonstrating that larger pathways do not distort the scores. (c) Heatmap analysis exposing scKGOT's internal mechanisms. (d) Sankey diagram providing a detailed visual representation of ligand-receptor-pathway connections, facilitating the identification and interpretation of key interactions.

### 4.3 Multi-level Analysis for Pathway Activation

Beyond achieving strong performance metrics, we now explore how scKGOT provides deeper, multi-dimensional insights into pathway activation within specific datasets. Using scKGOT, we conducted a comprehensive multi-level analysis of pathway activation from three perspectives: cells, genes, and pathways, as illustrated in Fig. 2. Each pair of cell types is treated as the fundamental unit of study, allowing us to systematically explore the signaling pathways and interactions specific to each cell type pair.

At the pathway level, we observe the distribution of pathway knowledge discrepancy scores (Fig. 2a), which reveals significant deviations from a normal distribution. This deviation

supports the robustness of scKGOT's predictions, demonstrating that our algorithm effectively leverages pathway information without being biased by the size of the pathways involved. In particular, larger pathways do not disproportionately influence the scores, as shown in Fig. 2b, highlighting the size-insensitive nature of scKGOT when identifying and ranking pathways.

Furthermore, our analysis extends to the internal mechanisms of scKGOT, where we perform a detailed co-analysis of ligand-receptor-pathway interactions (Fig. 2c). The heatmaps provide a multi-faceted visualization that captures the complexity of these interactions, offering deeper insights into how scKGOT integrates ligand-receptor interactions with pathway recognition. This co-analysis is crucial for understanding the underlying biological processes and for validating the model's predictive capabilities.

Finally, Fig. 2d presents a detailed visualization of the ligand-receptor-pathway co-analysis conducted using scKGOT. This Sankey diagram provides an intuitive representation of the connections between pathways, ligands, and receptors. The thickness of these connections reflects the strength of the predicted interactions, helping to prioritize key interactions for further biological investigation.

## 4.4 Pathway Interaction Analysis of Cell Type Pairs

Using scKGOT, we explored pathway interactions across three datasets with the goal of identifying critical gene interactions that might be overlooked in traditional analyses. By focusing on the largest connected components, we aimed to gain insights into how different cell types communicate within various biological contexts, contributing to a deeper understanding of processes such as immune modulation, cell signaling, and tissue remodeling.

In the placenta dataset (Fig. 3a), scKGOT identified interactions between lymphatic endothelial cells and villous cytotrophoblasts, highlighting pathways like TGF-$\beta$ signaling and integrin-ECM interactions (Heldin & Moustakas, 2016). These findings suggest that TGF-$\beta$ may play a role in immune tolerance and placental remodeling (Heldin & Moustakas, 2016), while integrin signaling appears important for maintaining placental structure. Clusters of immune-related genes also indicate that scKGOT could assist in predicting how the placenta manages inflammation and immune regulation.

In the testis dataset (Fig. 3b), scKGOT revealed significant interactions between Sertoli cells and spermatogonial stem cells, emphasizing the potential role of Wnt signaling in regulating spermatogenesis (Nusse & Clevers, 2017). Additionally, the identification of integrin-ECM genes suggests these interactions are key to the structural integrity of the seminiferous tubules (Lu et al., 2012). scKGOT further pointed to immune modulation, which may be crucial for protecting germ cells in the testis.

In the third dataset, comparing tumor and non-tumor liver sinusoidal endothelial cells interacting with pericytes, we observed distinct differences in the modules identified by scKGOT. In the tumor environment (Fig. 3c), ECM remodeling and Notch signaling pathways were notably present, suggesting their importance in promoting tumor progression (Lu et al., 2012). The tumor interactions revealed a more aggressive engagement with ECM components, likely facilitating cellular migration and invasion, while Notch signaling appeared to support the maintenance of tumor cell plasticity and the promotion of abnormal angiogenesis (Ferrara et al., 2003). Conversely, in the non-tumor condition (Fig. 3d), interactions seemed to be more focused on maintaining normal tissue structure and homeostasis. These findings underscore that the integration of LRP-KG with scRNA-seq data provides a powerful approach to generating biological hypotheses about cellular communication patterns in both normal and diseased states.

## 4.5 Ablation Study

To thoroughly evaluate the robustness and sensitivity of scKGOT, we designed a comprehensive ablation study consisting of three distinct experimental sets, each targeting specific aspects of the model (see Table 1 for details).

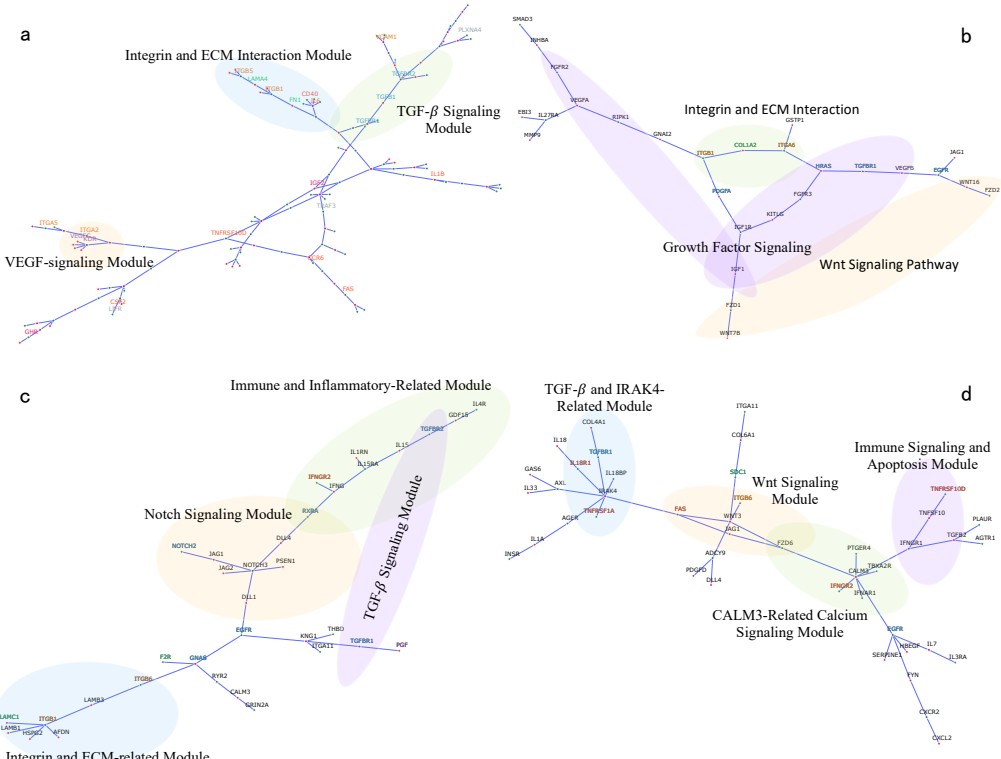

Figure 3: Visualization of the largest connected subgraphs for pathway interactions from three datasets. (a) Lymphatic endothelial cell and villous cytotrophoblast in the human placenta, highlighting TGF-$\beta$ and integrin-ECM interactions. (b) Sertoli cell and spermatogonial stem cell in the human testis, emphasizing Wnt signaling and immune modulation. (c) Tumor liver vascular endothelial cell and pericyte in human tumor, revealing ECM remodeling and Notch signaling driving tumor progression. (d) Non-tumor liver vascular endothelial cell and pericyte, showing distinct pathway interactions that maintain tissue structure and homeostasis. Ligands are represented by red circles, and receptors by green circles. Node colors correspond to functional categories: immune response (red-orange), cell signaling (blue), extracellular matrix (green), angiogenesis (purple), cell adhesion (orange), hormonal signaling (pink), neurotransmission (teal), and other functions (gray).

To assess the impact of reducing prior knowledge, we systematically removed facts (Fact-Drop) and pathway types (TypeDrop) from the LRP-KG. The results revealed that scKGOT maintained stable performance when up to 20% of pathway connections were removed. We believe this stability can be attributed to the inherent sparsity of the knowledge graph and the relatively limited amount of human-curated data. However, as the reduction increased to 30% or more, a noticeable decline in performance was observed, particularly in specific datasets. This suggests that there is a critical threshold beyond which the loss of information begins to significantly impair the model's ability to accurately predict ligand-receptor interactions. Additionally, when over 50% of pathway types were removed, scKGOT's results were markedly affected. This outcome is likely due to the model's ability to automatically identify and prioritize the most influential pathways, effectively disregarding less impactful routes even when substantial data reduction occurs.

To explore scKGOT's sensitivity to data reduction, we focused on the effects of removing less expressive genes (ExprDrop) and cells (CellDrop) from the scRNA-seq data. The findings showed that scKGOT sustained robust performance with up to a 30% reduction in under-expressed genes. However, beyond this point, performance deteriorated sharply, suggesting that the extensive removal of these genes disrupts the distribution of gene expression and diminishes the richness of potential information pathways, thereby reducing predictive

Table 1: Ablation Study of scKGOT performance on datasets with different sizes, tissues and species. Regarding robustness, we consider two aspects: prior knowledge and data availability. The prior knowledge part includes discarding facts (FactDrop) or pathway types (TypeDrop) by percentage from LRP-KG, while the data availability part includes dropping genes with low expression values (ExprDrop) or randomly removing cells (CellDrop) by percentage.

| | Human | | | | Mouse | | | |
|---|---|---|---|---|---|---|---|---|
| Tissue | $Pl_1$ | $Pl_2$ | $Tumor_1$ | $Tumor_2$ | $Brain_1$ | $Brain_2$ | Colon | MG |
| FactDrop 10% | 0.995 | 0.994 | 0.994 | 0.994 | 0.998 | 0.957 | 0.992 | 0.994 |
| FactDrop 30% | 0.983 | 0.836 | 0.993 | 0.994 | 0.998 | 0.931 | 0.991 | 0.994 |
| FactDrop 50% | 0.976 | 0.948 | 0.987 | 0.994 | – | – | 0.996 | 0.994 |
| FactDrop 70% | 0.937 | – | 0.982 | 0.993 | – | – | 0.975 | 0.992 |
| FactDrop 90% | 0.941 | 0.943 | – | – | – | – | – | 0.979 |
| TypeDrop 10% | 0.993 | 0.995 | 0.993 | 0.994 | 0.998 | 0.955 | 0.993 | 0.994 |
| TypeDrop 30% | 0.986 | 0.990 | 0.989 | 0.992 | 0.997 | 0.958 | 0.994 | 0.994 |
| TypeDrop 50% | 0.983 | 0.988 | 0.981 | 0.990 | 0.996 | 0.963 | 0.972 | 0.991 |
| TypeDrop 70% | 0.873 | – | 0.971 | 0.982 | – | – | – | 0.980 |
| TypeDrop 90% | – | – | – | – | – | – | – | 0.917 |
| ExprDrop 10% | 0.993 | 0.995 | 0.993 | 0.993 | 0.997 | 0.957 | 0.992 | 0.994 |
| ExprDrop 30% | 0.992 | 0.989 | 0.992 | 0.989 | 0.893 | 0.926 | 0.987 | 0.992 |
| ExprDrop 50% | 0.899 | 0.974 | 0.989 | 0.980 | 0.788 | 0.952 | – | 0.982 |
| ExprDrop 70% | 0.753 | 0.548 | 0.955 | 0.971 | 0.683 | 0.625 | – | 0.976 |
| ExprDrop 90% | – | – | – | – | – | – | – | – |
| CellDrop 10% | 0.991 | 0.995 | 0.994 | 0.993 | 0.998 | 0.958 | 0.992 | 0.995 |
| CellDrop 30% | 0.993 | 0.993 | 0.993 | 0.991 | 0.998 | 0.992 | 0.991 | – |
| CellDrop 50% | 0.991 | 0.994 | 0.987 | 0.989 | 0.998 | – | 0.992 | – |
| CellDrop 70% | 0.992 | 0.994 | 0.982 | 0.966 | 0.997 | 0.899 | 0.990 | – |
| CellDrop 90% | 0.990 | 0.988 | – | – | 0.996 | – | – | – |

accuracy. In contrast, scKGOT demonstrated considerable resilience to cell reduction, continuing to perform effectively even when up to 70% of cells were removed. We infer that this resilience is due to scKGOT's reliance on the mean expression levels of genes, allowing the model to derive accurate analyses from a smaller sample size without compromising the integrity of pathway predictions.

## 5 CONCLUSION

scKGOT integrates the Ligand-Receptor-Pathway Knowledge Graph (LRP-KG) with optimal transport to infer intercellular communication at the gene level, uncovering both known and novel biological pathways. It robustly maps the largest connected components in LRP-KG, which correspond to established pathways crucial for signal transduction in diverse cellular contexts. scKGOT's interpretability is enhanced by its visual flow-based mapping of gene interactions, enabling identification of regulatory nodes and therapeutic targets. The incorporation of optimal transport allows for fine-grained modeling of pathway dynamics, addressing variability in single-cell data and improving prediction accuracy of cell-cell interactions. This comprehensive framework not only advances pathway inference but also provides actionable insights for molecular biology and clinical applications by elucidating complex intercellular communication and offering potential for targeted therapeutic strategies.

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

## A    CRUCIAL PATHWAYS PER DATASET

This appendix presents a comprehensive list of crucial pathways identified by scKGOT for each dataset analyzed in our study. These pathways illuminate the intercellular communication patterns unique to different biological contexts, offering valuable insights into the underlying mechanisms of cell-cell interactions.

Researchers can reference this appendix to gain a deeper understanding of crucial pathways across various biological contexts and explore their implications for cellular processes and disease mechanisms.

| Tissue | Cell Summary | Pathway Summary |
|---|---|---|
| Brain | Endothelial cell
Neural stem cell | Progesterone-mediated oocyte maturation
ErbB signaling pathway
Melanogenesis |
| Brain | Endothelial cell
Neural stem cell | Th1 and Th2 cell differentiation
Non-alcoholic fatty liver disease (NAFLD)
Cell adhesion molecules (CAMs) |
| Brain | Neuron
Microglia | Viral protein interaction with cytokine and cytokine receptor
Neuroactive ligand-receptor interaction
Cytokine-cytokine receptor interaction |
| Colon | Neuron
Macrophage | Complement and coagulation cascades
Viral protein interaction with cytokine and cytokine receptor
NF-kappa B signaling pathway |
| Mammary Gland | Luminal cell
Basal cell | Signaling by Receptor Tyrosine Kinases
Longevity regulating pathway
Influenza A |
| Spinal Cord | Dividing myeloid cell
Astrocyte | Viral protein interaction with cytokine and cytokine receptor
Costimulation by the CD28 family
Th1 and Th2 cell differentiation |
| Spinal Cord | Dividing myeloid cell
Fibroblast | Cell adhesion molecules (CAMs)
Signaling by Receptor Tyrosine Kinases
Natural killer cell mediated cytotoxicity |

Table 2: Summary of predicted pathways across mouse tissues and cellular interactions, with three pathways per interaction due to space limitation.

| Tissue | Cell Summary | Pathway Summary |
|---|---|---|
| Embryo | Mitotic fetal germ cell
Meiotic fetal germ cell | Neuroactive ligand-receptor interaction
HIF-1 signaling pathway
Retrograde endocannabinoid signaling |
| Embryo | Mitotic fetal germ cell
Late granulosa | Complement and coagulation cascades
Cortisol synthesis and secretion
Regulation of lipolysis in adipocytes |
| Embryo | Meiotic fetal germ cell
Late granulosa | GABAergic synapse
VEGF signaling pathway
Dilated cardiomyopathy (DCM) |
| Embryo | Oogenesis fetal germ cell
Late granulosa | Non-alcoholic fatty liver disease (NAFLD)
TNF signaling pathway
Long-term potentiation |
| Embryo | Retinoid acid
signaling-responsive
fetal germ cell
Late granulosa | Breast cancer
Salmonella infection
Focal adhesion |
| Liver Bud | iPS cell-derived hepatic endoderm cell
iPS cell-derived endothelial cell | Viral protein interaction with cytokine and cytokine receptor
Neuroactive ligand-receptor interaction
GnRH secretion |
| Lung | Fibroblast
Alveolar type 2 cell | Progesterone-mediated oocyte maturation
Melanogenesis
Cell adhesion molecules |
| Placenta | Decidual natural killer cell 1
Extravillous trophoblast | Hedgehog signaling pathway
Viral protein interaction with cytokine and cytokine receptor
Complement and coagulation cascades |
| Placenta | Decidual natural killer cell 3
Extravillous trophoblast | Complement and coagulation cascades
Notch signaling pathway
GABAergic synapse |
| Placenta | Lymphatic endothelial cell
Syncytiotrophoblast | Hedgehog signaling pathway
GnRH secretion
Regulation of lipolysis in adipocytes |
| Placenta | Lymphatic endothelial cell
Villous cytotrophoblast | Renin secretion
Long-term potentiation
Hemostasis |
| Testis | Sertoli cell
Spermatogonial stem cell | Fc epsilon RI signaling pathway
VEGF signaling pathway
Signaling pathways regulating pluripotency of stem cells |
| Tumor | Liver sinusoidal endothelial cell
Pericyte | ECM remodeling
Notch signaling
Angiogenesis |
| Tumor | Tumor liver vascular endothelial cell
Pericyte | Viral protein interaction with cytokine and cytokine receptor
Basal cell carcinoma
Adipocytokine signaling pathway |

Table 3: Summary of predicted pathways across human tissues and cellular interactions

## B    LIMITATION

When working with a limited number of cells, the statistical robustness of observations can be significantly compromised due to small sample sizes, leading to ill-defined problems. In such scenarios, many genes may appear unexpressed simply because the sample is not adequately representative of the true biological diversity. This sparsity can introduce biases, resulting in misleading interpretations of signaling pathways and gene interactions. Moreover, the reliability of the inferred intercellular communication networks diminishes, as sparse data may fail to capture the full complexity of the underlying biological processes. To mitigate these challenges, it is crucial to ensure sufficient cell coverage and consider complementary methods or datasets to validate the findings, thereby enhancing the reliability and interpretability of the results.

## C    LIGAND-RECEPTOR-PATHWAY CO-ANALYSIS

Fig. 4 is divided into three main sections from top to bottom: pathway-ligand, ligand-receptor, and receptor-pathway, derived from human placenta data with 20,218 cells. The sender cell type is lymphatic endothelial cells, while the receiver cell type is villous cytotrophoblasts. The color-coded edges represent connections within the same pathway, enabling easy identification of predominant pathways across the predicted results. Intersections between different pathways also highlight ligand-receptor pairs associated with multiple pathways, showcasing cross-pathway communication, one of the key insights revealed by the scKGOT algorithm.

In the central ligand-receptor section, the thickness of the lines corresponds to the predicted interaction strength, helping to identify the top-ranked pairs among numerous connections. For example, the IGF1-IGF1R pair is highly ranked and integrates signals from various pathways, including pathways in cancer, glioma, HIF-1 signaling, oocyte meiosis, mTOR signaling, and others. The significant thickness of the line connecting IGF1 and IGF1R suggests a strong interaction prediction, warranting further investigation. Note that this Sankey diagram is a more complete version of Fig. 2f.

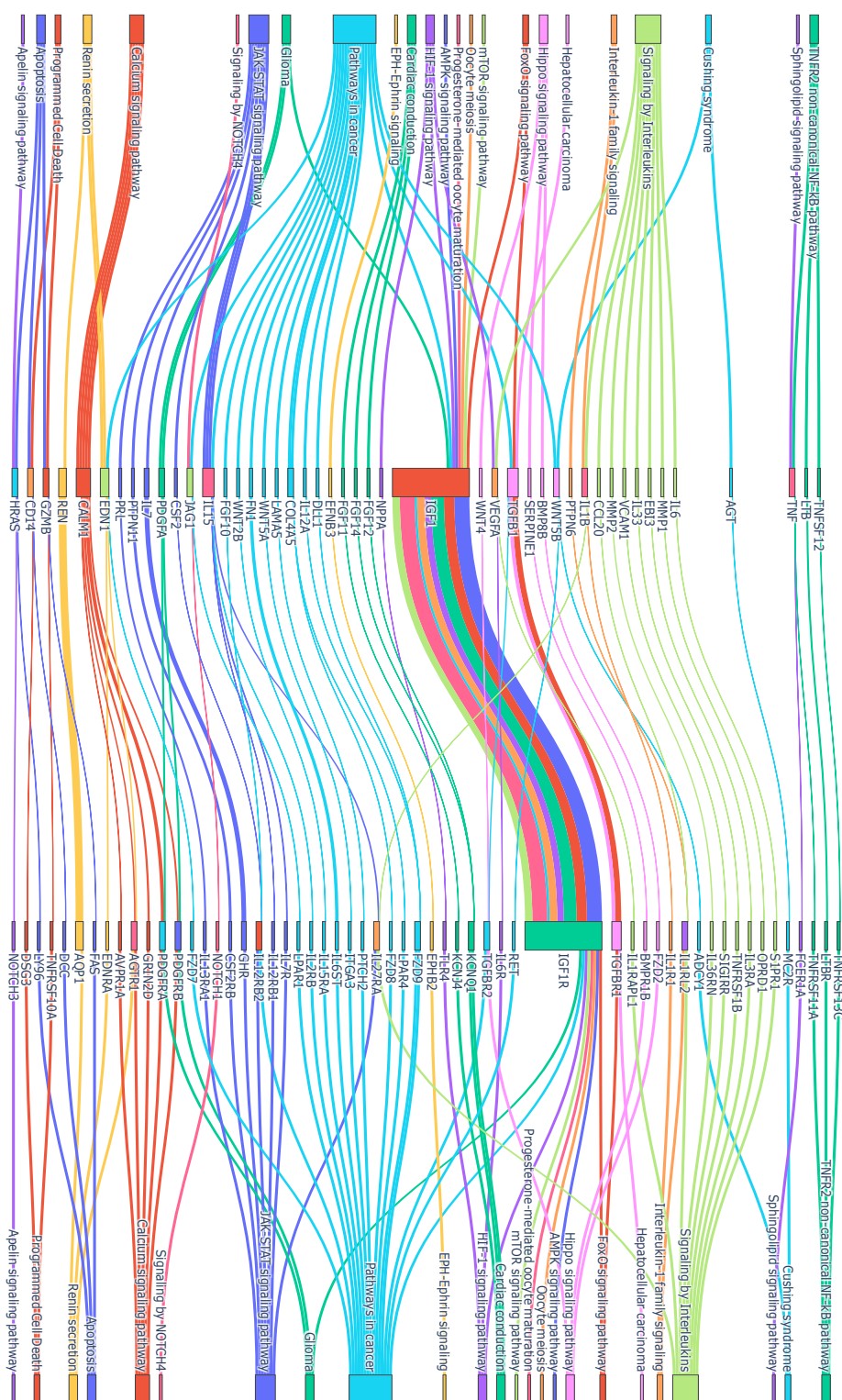

Figure 4: Sankey diagram illustrating the predicted interactions, with sections from the top to bottom representing pathway-ligand, ligand-receptor, and receptor-pathway. The color-coded edges indicate connections within the same pathway. In the ligand-receptor section, line thickness reflects the strength of the predicted interactions.

