# OpenReview forum: "scKGOT: Intercellular Signaling Inference with Knowledge Graph Optimal Transport for Single-cell Transcriptomics"
_ICLR.cc/2025/Conference — Submitted to ICLR 2025_

### Official Review · Reviewer_krvi · 2024-11-01

**Soundness:** 2
**Presentation:** 2
**Contribution:** 2
**Rating:** 3
**Confidence:** 3

**Summary:**

The paper introduces scKGOT, a method that uses Knowledge Graph Optimal Transport (KGOT) to infer intercellular signaling pathways based on single-cell RNA sequencing data. The method aims to improve ligand-receptor signaling network inference by integrating knowledge from a Ligand-Receptor-Pathway Knowledge Graph (LRP-KG). scKGOT dynamically models ligand-receptor relationships between cells and identifies active pathways by optimizing transport based on gene expression profiles and prior pathway information. This approach is benchmarked against existing methods and is claimed to offer better precision and interpretability in cell signaling pathway analysis.

**Strengths:**

1. Innovative Approach: Utilizes optimal transport within a knowledge graph framework, which is novel and relevant to understanding cell-cell communications.
2. Integration of Pathway Knowledge: Incorporates LRP-KG as a prior, which is a valuable addition to improve biological relevance
3. Intepretability: The model offers insights at various biological levels (cells, genes, pathways), useful for specific cell-type comparisons and disease studies.

**Weaknesses:**

1. Limited Clarity on Dynamic Adaptability: While the paper claims that Optimal Transport allows scKGOT to dynamically adapt to different biological conditions, it lacks clear explanations and quantitative examples demonstrating how this dynamic adaptability outperforms static models in specific biological scenarios.
2. Dependence on Complete Pathway Information: scKGOT relies on a comprehensive Ligand-Receptor-Pathway Knowledge Graph (LRP-KG) as a prior, which may not be fully available or accurate for all biological contexts. This dependency raises questions about the model’s robustness and effectiveness in datasets with incomplete or biased pathway annotations.
3. Complexity and Accessibility: The use of Optimal Transport, combined with pathway knowledge integration, makes scKGOT computationally complex and potentially challenging for researchers without advanced computational expertise. This complexity could limit its accessibility and reproducibility.

**Questions:**

Majors:
1. You mention that the Optimal Transport (OT) method in scKGOT captures more dynamic signaling patterns. Specifically, could you explain how OT dynamically adjusts under different biological conditions? Could you quantitatively demonstrate its performance over static probability models, like CellChat, on particular datasets?
2. Regarding the use of the Ligand-Receptor-Pathway Knowledge Graph (LRP-KG), how does scKGOT handle cases where pathway information is incomplete or biased? In scenarios with limited prior knowledge, does the algorithm experience significant performance limitations or weaker results on specific datasets?
3. The paper mentions that scKGOT’s dynamic adaptability can handle heterogeneous data. Could you provide examples of how this adaptability performs in specific applications (e.g., complex tumor microenvironments or disease states)? Additionally, was this heterogeneity validated in real-world tests?
4. Considering the computational complexity of the OT algorithm, how efficient is scKGOT on large-scale single-cell datasets? Has there been a quantitative comparison of speed and scalability on large datasets against simpler models, like CellChat?

Minors:
1.	Check your grammar, for example, section 3 line 3: “we defines …”
2.	Figure 1 legend, what does color represent? Is scKGOT better than others?

---

### Official Review · Reviewer_mhe5 · 2024-11-04

**Soundness:** 1
**Presentation:** 1
**Contribution:** 1
**Rating:** 1
**Confidence:** 5

**Summary:**

scKGOT uses optimal transport to solve a multi-relation link prediction problem to predict likely ligand pathway-receptor pathway connections using single cell transcriptomics data.

**Strengths:**

Most cell-cell interaction prediction methods solely predict specific and direct ligand-receptor interactions. This method extends the prediction to pathways on both sides - both in the sender and receiver cells.

**Weaknesses:**

It is very difficult to understand the rationale and evaluation methods used in this paper. On the one hand, the authors make a good point that pathways in sender and receiver cells are related via causal links (i.e. the pathway in the sender cell creates a ligand that get secreted and physically interacts with a receptor on a receiver cell, which causes a pathway to activate within the receiver cell). On the other hand, the main problem I have is that there is no ground truth for this type of relationship presented, just a general comparison of learned networks and prior networks. Further much of the presented biology makes very little sense. For example, in figure 2d, there is a relationship where "pathways in cancer" generates a large number of ligands, which bind to a large number of receptors in another "pathways in cancer" pathway. This really makes not sense biologically and I cannot understand how it makes sense to predict it as relevant.

Another example "By focusing on the largest connected components, we aimed to gain insights into how different cell types communicate within various biological contexts, contributing to a deeper under- standing of processes such as immune modulation, cell signaling, and tissue remodeling." - why is the largest connected component interesting here? The networks in Figure 3 almost look like a random assortment of pathways and don't provide any cell context information, which seriously undermines their use as a tool to interpret the results of scKGOT.

Multiple statements in the paper demonstrate poor understanding of the biological application area e.g. "2,223,641 and 1,651,421 records of ligand-receptor interaction facts respectively, including binding, dephosphorylation and activation, etc." - these are mostly not direct ligand-receptor relationships, but rather are relationships within pathways.

**Questions:**

How well would using a random knowledge graph perform in the evaluation presented?

---

### Official Review · Reviewer_RU5G · 2024-11-08

**Soundness:** 2
**Presentation:** 2
**Contribution:** 3
**Rating:** 3
**Confidence:** 4

**Summary:**

This paper introduces scKGOT (single-cell Knowledge Graph Optimal Transport), a computational method for analyzing cell-to-cell communication in single-cell transcriptomic data using a customized optimal transport algorithm. The method combines a Knowledge Graph Optimal Transport algorithm with prior biological knowledge through a Ligand-Receptor-Pathway Knowledge Graph (LRP-KG) to model and quantify signaling networks between sender and receiver cells. The optimal transport model in scKGOT is used to find the optimal way to "transport" signals between sender and receiver cells through signaling pathways. The model incorporates prior knowledge from LRP-KG as initial guidance but allows for dynamic adaptation based on gene expression data. This helps identify both known and potentially novel signaling pathways between cells.

**Strengths:**

The idea of incorporating prior biological knowledge (in this paper the LRP knowledge graph) to discover cell-cell interactions in scRNA-seq data definitely brings another layer of enhanced interpretability to the existing developed model for cell-cell interaction inference and it has novelty. Additionally, the usage of optimal transport in scKGOT  to find the optimal way to "transport" signals between sender and receiver cells through signaling pathways is potentially novel.

**Weaknesses:**

1- Literature review in the introduction section needs to be improved: For example, the introduction section is very limited in reviewing previous works that utilized prior biological pathway knowledge to enhance model interpretability in the field of omics data analysis. The authors may need to refer to some previous works such as Spatalk [1], EXPORT [2], VEGA[3] that utilize knowledge graphs to enhance model interpretability when analyzing omics data.

2- Mathematical description of paper needs to be significantly improved: Some mathematical notations are not described at all when describing the proposed model. For example, what are s1 and s2 in equation (2)? or a and b in equation (3)?  also what's the intuition behind inequalities in (3)? Major revision is needed in mathematical description of the model to help readers better understand the model

3- The results need to be better described:

--For example whats the difference between left and right figures in Figure 1 (top row)? What does different colors mean? Also what does (CTDB) version of benchmarking tools mean in Figure 1 (bottom row) mean? Can authors provide more descriptive caption for figure 1?

--In Figure 2C, why authors have focused on "Eph-Ephrin Signaling" or "Signaling By Notch4" pathways? Do they have any specific characteristics that authors have picked them for visualization? What does each subfigure indicate?

-- In Figure 2D, can authors focus on one specific LRP pair that is previously well studied and describe the model results on that specific LRP pair? Its hard to see what result is novel from scKGOT in Figure 2D.

References:
[1] Shao, X., Li, C., Yang, H. et al. Knowledge-graph-based cell-cell communication inference for spatially resolved transcriptomic data with SpaTalk. Nat Commun 13, 4429 (2022). https://doi.org/10.1038/s41467-022-32111-8
[2] Biologically Interpretable VAE with Supervision for Transcriptomics Data Under Ordinal Perturbations
Seyednami Niyakan, Byung-Jun Yoon, Xiaoning Qian, Xihaier Luo bioRxiv doi:https://doi.org/10.1101/2024.03.28.587231
[3] Seninge, L., Anastopoulos, I., Ding, H. et al. VEGA is an interpretable generative model for inferring biological network activity in single-cell transcriptomics. Nat Commun 12, 5684 (2021). https://doi.org/10.1038/s41467-021-26017-0

**Questions:**

The authors need to enhance literature review quality as well as results section and mathematical model formulation to help readers understand the model novelty as well as its imitations.

---

### Official Review · Reviewer_m1uf · 2024-11-09

**Soundness:** 3
**Presentation:** 2
**Contribution:** 2
**Rating:** 3
**Confidence:** 4

**Summary:**

In this manuscript, the authors present scKGOT, a novel computational method that leverages Knowledge Graph Optimal Transport algorithms to infer cell-cell communication networks from single-cell RNA sequencing data. The method combines prior biological knowledge from a Ligand-Receptor-Pathway Knowledge Graph with dynamic adaptation to gene expression data, demonstrating superior precision and interpretability compared to existing approaches across multiple biological case studies. Specifically, scKGOT operates in three key steps: (1) it utilizes a knowledge graph (LRP-KG) containing known gene-gene interactions and pathway information, (2) it employs a novel algorithm incorporating gene importance metrics and pathway Knowledge Discrepancy to identify activated signaling pathways and confident ligand-receptor pairs, and (3) it reconstructs intercellular signaling networks by combining its predictions with established biological knowledge.

**Strengths:**

The paper demonstrates strong originality through several key aspects. At its core, it presents a novel problem formulation that reframes cell-cell communication as an optimal transport problem integrated with knowledge graphs. This approach moves beyond simple binary predictions to introduce comprehensive pathway-level analysis, supported by an innovative scoring framework that combines gene importance with pathway knowledge discrepancy. It is impressive how the authors merge three distinct approaches, optimal transport theory, knowledge graph embeddings, and single-cell transcriptomics analysis while successfully integrating prior biological knowledge with data-driven insights. The paper brings new perspective to biological pathway analysis through its novel application of optimal transport, effectively bridging machine learning and systems biology while introducing a new framework for analyzing complex cellular interactions.
The quality of the work is particularly strong. The paper builds on a solid mathematical foundation, employs a comprehensive evaluation framework, and includes thorough ablation studies that demonstrate robust performance across multiple datasets.
In terms of clarity, the paper succeeds in presenting complex ideas in an accessible manner. The organization follows a logical flow of ideas, with a well-structured methodology section and clear presentation of results. The visualizations are particularly effective, employing comprehensive tools and clear performance comparisons.
Its scientific impact is evident in how it advances our understanding of cell-cell communication, provides new tools for biological pathway analysis, and could potentially impact disease understanding and drug development. The methodological contribution is equally important, creating a new framework for analyzing complex biological systems while demonstrating successful integration of domain knowledge with machine learning.
The visualization of pathways using Sankey diagrams is particularly impressive, effectively illustrating the complex predicted interactions across pathways, ligands, and receptors. These diagrams not only help in understanding the flow of signals but also in identifying key interaction hubs and dominant pathways, making complex biological information more accessible and interpretable.

**Weaknesses:**

While scKGOT presents an innovative and mathematically sound approach for analyzing cell-cell communication, the manuscript would significantly benefit from more detailed methodological descriptions. The current presentation leaves several critical implementation questions unanswered. Specifically, the paper should elaborate on the preprocessing pipeline for single-cell transcriptomics data, including how it handles common technical challenges like dropout effects, batch variations, and cell type annotation reliability.
A crucial aspect that requires more clarity is the construction and quality control of the knowledge graph from multiple pathway databases. While the authors utilize both KEGG and Reactome databases (with over 2 million interaction records), the methodology for resolving redundancies and potentially conflicting information between these databases remains unclear. The integration strategy for different database formats, confidence scores, and annotation systems would be valuable information for readers looking to implement or build upon this approach and can be added as supplementary information.
Minor Comments:
Writing: There is redundant information in the introduction section (paragraphs 3 and 4 are repetitive)
Implementation: Additional details about computational requirements and scalability would be valuable
Method availability: Information about code availability and documentation would benefit the community

**Questions:**

1. Could the authors provide detailed methodology for preprocessing of single-cell data (normalization, filtering, batch size effects etc)
This information is crucial for reproducibility and understanding method limitations.
2. Could the authors comment on the computational complexities of different data sizes and if there is any minimum/maximum data size recommendations to implement this method
3. How were the pathways integrated from different sources and if any scoring method was used to handle redundant information.
4. Could the authors  provide more details about the parameter selection process, particularly: Thresholds for gene importance scores, Criteria for pathway knowledge discrepancy. How these parameters might need adjustment for different dataset sizes?

---

### Meta-Review · Area_Chair_6XmB · 2024-12-19

**Metareview:**

This paper proposes scKGOT, a novel method that employs the KGOT (Knowledge Graph Optimal Transport) to model the ligand-receptor-signaling networks between sender and receiver cells.
The reviewers recognize the novelty of the proposed method as well as the importance of incorporating prior knowledge into the analysis.
However, there are significant concerns regarding the literary and technical presentations of the work, unclear rationale/motivation of the work, lacking clarity regarding the methodology and results, and insufficient discussion of relevant literature.

**Additional Comments On Reviewer Discussion:**

The authors have not responded to the reviewers' concerns during the discussion period and none of the concerns/comments have been addressed.

---

### Decision · Program_Chairs · 2025-01-22

Reject